# Is maternal autonomy associated with child nutritional status? Evidence from a cross-sectional study in India

**Pintu Paul**[1]*, **Ria Saha**[2]

1 Centre for the Study of Regional Development, School of Social Sciences, Jawaharlal Nehru University, New Delhi, India, 2 Senior Public Health Intelligence Analyst, Medway Council, Chatham, England

☯ These authors contributed equally to this work.

* pintupaul383@gmail.com

**Data Availability Statement:** Data used in this study is publicly available and can be accessed from the DHS program website at https://dhsprogram.com/methodology/survey/survey-display-355.cfm.

## Abstract

Despite India's steady economic growth over recent the period, the burden of childhood malnutrition persists, contributing to higher neonatal and infant mortality. There is limited evidence available to contextualise mothers' crucial role in childcare practices and health status in the Indian context. This study attempts to assess the association between maternal autonomy and the nutritional status of children under five. We used samples of 38,685 mother-child pairs from the fourth round of the National Family Health Survey (NFHS-4), conducted in 2015–16. We considered three widely used indicators of child nutrition as outcome variables: stunting, wasting, and underweight. Maternal autonomy (measured from three dimensions: household decision-making, freedom of physical movement, and access to economic resources/control over assets) was the key predictor variable, and various child demographics, maternal, and household characteristics were considered control variables. Stepwise binary logistic regression models were performed to examine the association. Of study participants, 38%, 21%, and 35% of children were stunted, wasted, and underweight, respectively. Our results (models 1 to 4) indicate that mothers with greater autonomy were significantly associated with lower odds of malnourished children. After controlling for all potential confounding variables (in model 5), maternal autonomy had a statistically insignificant association with children's stunting (Odds ratio [OR]: 0.93; 95% confidence interval [CI]: 0.87, 1.00) and wasting (OR: 0.92; 95% CI: 0.85, 1.00). However, a significant relationship (though marginally) was retained with underweight (OR: 0.94; 95% CI: 0.88, 0.99). In addition, socio-demographic characteristics such as child age, birth order, maternal education, maternal BMI, place of residence and household wealth quintile were found to be strong predictors of child nutritional status. Future policies should not only inform women's empowerment programmes but also emphasise effective interventions toward improving female educational attainment and nutritional status of women, as well as addressing socio-economic inequalities in order to combat the persistent burden of childhood malnutrition in India.

**Funding:** The author(s) received no specific funding for this work.

**Competing interests:** The authors have declared that no competing interests exist.

## Introduction

Childhood malnutrition remains a major public health problem in low- and middle-income countries (LMICs). Malnutrition contributes to nearly half of all deaths in children below five years of age [1, 2]. It increases the frequency and severity of infections, putting children at risk of dying from diseases that are easily preventable [1, 2]. Moreover, undernourishment during infancy has a detrimental effect on health and cognitive development in adulthood [3, 4]. Several studies have identified socioeconomic, demographic, and household environmental factors that influence children's nutritional status [1, 5, 6]. Children's malnutrition is caused by a wide variety of factors, such as poor diet, illnesses (e.g., malaria and water-borne diseases), limited access to clean water and sanitation facilities, unsafe hygiene practices, lack of access to health services, and insufficient or inappropriate child feeding practices [1, 7, 8]. Although India has made remarkable achievements in economic growth over the past years, the burden of childhood malnutrition continues to remain alarmingly high [9]. The latest National Family Health Survey (NFHS-5) of India indicated that 36%, 19%, and 32% of children were stunted, wasted, and underweight, respectively, in 2019–21, a slight reduction from NFHS-4 (2015–16) [10]. India's persistent prevalence of malnutrition leads to a higher incidence of preventable deaths among children, especially in high burden states such as Bihar, Uttar Pradesh, Madhya Pradesh, Chhattisgarh, Jharkhand, and Orissa [9, 11, 12]. In 2017, malnutrition accounted for 68% of the total under-five deaths in India [9].

Among several maternal factors, maternal autonomy emerges as a crucial predictor of the nutritional status of children. Women's autonomy is a multi-dimensional and context-specific aspect, diversely defined in the existing feminist literature. Kabeer [13] has put forward the notion of women's empowerment as the ability to exercise strategic life choices in terms of resources, agency, and achievements. Dyson and Moore [14] defined female autonomy as "the ability. . . to obtain information and to use it as the basis for making decisions about one's private concerns and those of one's intimates." Jejeebhoy and Sathar [15] described autonomy as "the control women have over their own lives—the extent to which they have an equal voice with their husbands in matters affecting themselves and their families, control over material and other resources, access to knowledge and information, the authority to make independent decisions, freedom from constraints on physical mobility, and the ability to forge equitable power relationships within families."

Most of the previous studies conducted in South Asian countries (including India) have found a positive relationship between maternal autonomy and child nutritional status. It is evident that the degree of association substantially varies across various dimensions of women's autonomy. Smith et al. [16] using cross-country data in three developing regions (South Asia, Sub-Saharan Africa, and Latin America and the Caribbean) found that women's status–*women's power relative to men*–has a significant, positive influence on the nutrition status of children under three years of age [16]. While the study found that women's higher status positively impacts children's short and long-term nutritional status in South Asia and Sub-Saharan Africa, the pattern is different in Latin American and Caribbean countries where women's higher social status was found to be linked with short-term improvement of children's nutritional status concentrated in households where women hold low decision-making power [16]. Most of the existing studies carried out in different sub-national regions of India also found a positive linkage between maternal autonomy and child nutritional status. For instance, a study in Andhra Pradesh, India using NFHS-2 data (1998–99) showed that mothers' financial autonomy and physical mobility are significantly associated with reduced odds of child stunting after controlling for child's sex, standard of living, place of residence, and mother's education [17]. In another study, Shroff et al. [18] reaffirmed that mothers with higher

participation in household decisions are less likely to have underweight and wasted infants in rural Andhra Pradesh, India. Sethuraman et al. [19] in a study of rural Karnataka, India demonstrated that while mothers' position in households and involvement in decision-making and their physical mobility within the village improved children's nutritional status, experience of domestic violence (psychological abuse and sexual coercion) increased the risk of childhood malnutrition. A community-based study conducted among slum children in Malda district, India, found a positive association between various dimensions of maternal autonomy and child nutritional indicators to some extent [20]. Authors found that mothers' financial independence and household decision-making are positively related to children's z-score for height-for-age (HAZ) and weight-for-height (WHZ), respectively, even after adjusting for age at pregnancy, maternal BMI, and education [20]. However, a few studies have provided evidence of an association between maternal autonomy and childhood nutrition using nationwide data and found inconsistent findings [21, 22]. This inclusive evidence is insufficient for the effective designing of evidence-informed policies and interventions to combat the burden of persistent childhood malnutrition in India. Studies conducted in Pakistan [23], Nepal [24], and Bangladesh [25–27] highlight that maternal autonomy has a strong association with the nutritional status of children, where greater maternal empowerment could potentially reduce the risk of childhood malnutrition. In Pakistan, mothers' higher levels of education and employment status enable them to directly engage in household decision-making and buy high-quality dietary food for children, which is positively related to the nutritional status of children [23]. Similarly, a cross-sectional study from the rural Kaski district of Nepal documented that intrinsic factors such as maternal education and community membership associated with maternal empowerment reduced the likelihood of having underweight and stunted children [28]. In Bangladesh, Rahman et al. [26] found that mothers' participation in household decision-making is associated with lower odds of childhood malnutrition.

Evidence from African countries shows a weak and inconsistent relationship between women's autonomy and childhood malnutrition. A multi-country study in Sub-Saharan Africa shows that while women's justification of wife-beating and experience of violence increased the likelihood of child stunting, involvement in decision-making has a negative relationship with it [29]. In Malawi, Chilinda et al. [30] suggest a weak and marginal association between maternal autonomy and child stunting, where the odds of stunting were reduced with higher levels of maternal autonomy even after controlling for the child's age, sex, mother's age, and maternal BMI. However, the association was no longer significant after adjusting for maternal education, household wealth, and area of residence in the analysis. A study conducted in Lao PDR among semi-urban communities found that women's self-efficacy for health services, self-esteem, and control over money significantly reduce the odds of childhood stunting; however, decision-making power and freedom of mobility have no significant relationship with children's stunting [31].

It is pertinent to understand the pathways through which maternal autonomy influences child growth and nutritional status. The United Nations Children's Fund (UNICEF) underlines optimal childcare practices as an important aspect in the prevention of malnutrition among infants and under-five children [32]. As primary caregivers, mothers' control over material resources, access to finance, and participation in household decisions are critical for the health and well-being of children [33]. Autonomy of mothers enables them to allocate (financial) resources for their children's health-related expenditures, including buying the most nutritious supplements. Evidence also shows a positive link between maternal autonomy and patterns of child feeding practices [34]. Moreover, women with greater autonomy are more likely to utilise essential maternity care services [35, 36]. Maternal health and nutritional

status (e.g., BMI of mothers) are potentially linked to neonatal birth outcomes and have implications for early childhood growth and development [37, 38].

Although a growing body of literature has examined the influence of maternal autonomy on child growth and nutrition, the association remains inconsistent and inclusive. Most of the previous studies in India have been carried out in small study settings, which provide insights into the role of maternal autonomy on child nutrition in a particular regional context, and therefore nationally representative evidence is scarce. Given the pervasiveness of childhood malnutrition in India and the limited evidence on the influence of maternal autonomy on child nutrition to date, it is crucial to unravel the role of maternal autonomy in child nutritional outcomes for evidence-based policymaking toward the effective reduction of childhood malnutrition. Against this backdrop, we undertook a comprehensive study to examine the association between maternal autonomy (assessed through three domains of autonomy: household decision-making, freedom of movement, and access to financial resources) and the nutritional status of children under five using a nationally representative population-based survey [39].

## Materials and methods

### Data source

This study used data from the fourth round of the National Family Health Survey (NFHS-4), conducted in 2015–2016 [39]. It is a large-scale, nationally representative, population-based survey, covering all states and union territories of India. The survey was conducted by the International Institute for Population Sciences (IIPS) under the supervision of the Ministry of Health and Family Welfare (MoHFW), Government of India [39]. The NFHS-4 interviewed 601,509 households with a response rate of 98% and 699,686 women aged 15–49 years with a response rate of 97%. The survey provides essential information on various aspects of population, health, and family welfare, such as fertility, mortality, maternity care utilization, family planning, child health and nutrition, non-communicable diseases, women's autonomy, and domestic violence. In this survey, a two-stage stratified sampling design was adopted for the selection of the participants. In total, 28,586 clusters (primary sampling units) were chosen, of which fieldwork was completed for 28,522 clusters. The 2011 Census enumeration served as the sampling frame for the selection of clusters. In the first stage, the clusters were selected using probability proportional to size (PPS). In the second stage, complete household mapping and listing operations were carried out in the selected clusters, and 22 households were randomly chosen in each cluster from the household listing. A detailed description of the sampling design and survey procedure is provided in the NFHS-4 national report [39]. In the present study, we utilised eligible women's information on autonomy, children's biomarker information, and background characteristics of women and children from a subset of households that are only selected for the state module.

### Study participants

A total of 259,627 children under five were interviewed by the NFHS-4. Women's autonomy-related information was collected only for the state module, which comprised 15% of the total eligible women aged 15–49 years in the survey. A total of 45,231 mother-child pairs were available in the module. Among them, anthropometric information was collected for 41,158 children, of which 184 children's anthropometric measurements were out of plausible limits and 1749 were flagged cases. Therefore, 39,225 children had valid information on anthropometric measurements. Since only currently married women were asked questions on household decision-making participation, we excluded 540 women who were not currently married

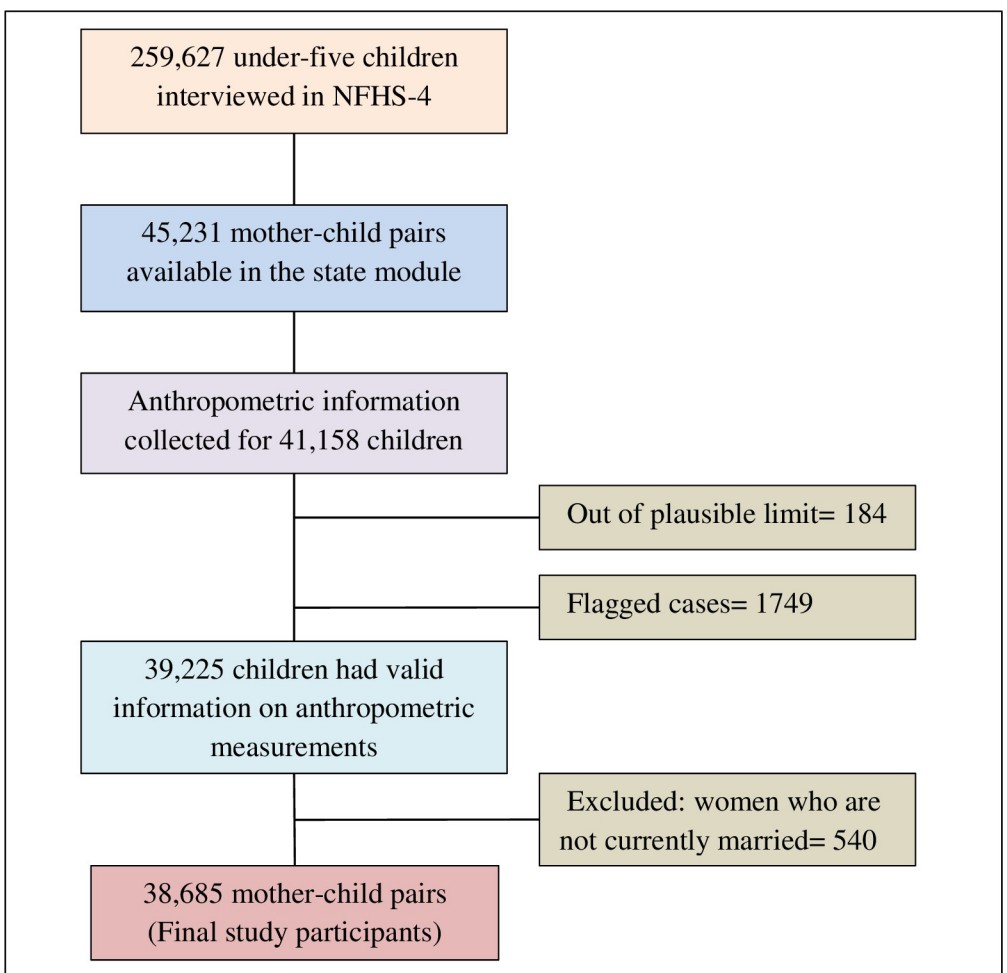

**Fig 1. Flow chart showing the selection of study participants, NFHS-4 (2015–16).**

(widowed/divorced/separated/not living together) from the study sample. Therefore, 38,685 mother-child pairs constitute the final study participants in this study (**Fig 1**).

## Outcome variables

Children's nutritional status was the outcome of interest in this study. The nutritional status of children was assessed from anthropometric measurements (e.g., height and weight) collected during the survey. In this study, we considered three widely used anthropometric indices: stunting, wasting, and underweight. Stunting is a symptom of chronic or recurring undernutrition, which is caused by long-term nutritional deprivation, reflecting poor socioeconomic and household environmental conditions [40]. Wasting indicates acute malnutrition, which is typically caused by insufficient food intake or a high incidence of infectious diseases (e.g., diarrhoea), resulting in severe weight loss [40]. Underweight is a combination of both chronic and acute malnourished conditions [40]. Children with z-scores less than two standard deviations from the WHO child growth standards for height-for-age (HAZ), weight-for-height (WHZ), and weight-for-age (WAZ) were classified as stunted, wasted, and underweight, respectively [41].

## Key predictor

Women's autonomy was the key predictor variable in this study. We considered three dimensions of women's autonomy in the analysis: (1) household decision-making, (2) freedom of movement, and (3) access to economic resources/control over financial assets. These three dimensions have been widely used as a measure of women's autonomy/empowerment in the existing literature [15, 42, 43].

Decision-making autonomy was measured across three areas of household decisions: (a) making major household purchases, (b) own healthcare, and (c) visiting family or relatives. The responses of participants were recorded as respondent, husband, respondent and husband jointly, someone else, and others. Each question was dichotomized where women who participated in household decisions were coded as "1," otherwise coded as "0."

Freedom of movement was assessed by women's physical mobility to the following three places: (a) market, (b) health facility, and (c) places outside this village/community. The responses were recorded as being alone, with someone else, and not at all. Three separate dichotomized variables were created where women who were allowed to go alone to these places were coded as "1", otherwise coded as "0."

Women's access to economic resources was assessed using three questions. Women enquired whether they had the following: (a) a bank or savings account; (b) owned a house either alone or jointly with someone else; and (c) owned any land either alone or jointly with someone else. All three items were dichotomized if women who own these financial assets were coded as "1," otherwise coded as "0."

We constructed the women's autonomy index (WAI) by combining all nine items corresponding to three domains of autonomy. The score ranges from 0 to 9. Women were categorised into three groups based on the scores derived from the WAI: low autonomy (0–2), moderate autonomy (3–6), and high autonomy (7–9).

## Confounding variables

We considered several confounding variables in the analysis to assess the association between maternal autonomy and child nutrition. These variables include child demographics, maternal characteristics, and household characteristics. Children's demographics include their age (0–11, 12–35, and 36–59 months), gender (male and female), and birth order (<3 and 3+). Maternal characteristics include maternal age (15–24, 25–34, and 35–49 years), educational attainment (no education, primary, secondary, and higher education), and maternal BMI (categorized as per WHO cut-off points: <18.5 kg/m2 [underweight] 18.5–24.9 kg/m2 [normal], and ≥25.0 kg/m2 [overweight/obese]). Household characteristics include the place of residence (rural and urban), caste (Scheduled Caste [SC], Scheduled Tribe [ST], Other Backward Class [OBC], and none of them [forward caste]), religion (Hindu, Muslim, and other religions), household size (0–4, 5–6, and 6+), sex of the household head (male and female), and household wealth quintile (poorest, poorer, middle, richer, and richest).

## Statistical analysis

We performed descriptive statistics to show the distribution of study participants by various characteristics. We estimated the percentage distribution of nutritional outcomes (stunting, wasting, and underweight) by the key predictor and confounding variables. Pearson's chi-square statistic was used to test the differences in nutritional status by the selected explanatory variables. A series of stepwise binary logistic regression models were employed to assess the association between women's autonomy and child nutritional status. Five regression models were employed for each nutritional indicator. Model 1 shows the bivariate (crude) association

between maternal autonomy and child nutrition. In model 2, child demographics were included. We progressively added maternal characteristics in model 3. Model 4 comprises maternal autonomy, child demographics, and household characteristics. In the full model (model 5), all confounding variables (child demographics, maternal, and household characteristics) were controlled. We checked for multicollinearity between the independent variables using variation inflation factors (VIF: 1.66) and found no evidence of collinearity in the analysis. The results of logistic regression models were presented in odds ratio (OR) with a 95% confidence interval (CI). The appropriate sample weight was used to estimate the results. All statistical analyses were performed using STATA version 14.0 (StataCorp LP, College Station, TX, USA).

### Ethics statement

The ethical approval of the NFHS-4 (2015–16) was obtained from the ethics review board of the International Institute for Population Sciences (IIPS), Mumbai. The survey was also reviewed and approved by the ICF International Review Board (IRB). For participation in this survey, informed written consent was obtained from the respondents during the survey. Each participant's approval was sought, and then only interviews were conducted. The NFHS-4 is an anonymous, publicly available dataset with no identifiable information about the survey participants and is accessible upon a granted request from the Demographic and Health Surveys (DHS) Program at https://dhsprogram.com/data/available-datasets.cfm.

## Results

### Characteristics of study participants

Of the study sample (n = 38,685), the majority of women had a moderate (58%) to a high level (23%) of autonomy. About one in every five children was an infant (mean = 29.9 months; SD = 17). The share of male and female children was almost equal in the study sample (51% vs. 49%). The majority (70.7%) were either first- or second-order children. About one-third of mothers (34.3%) were in the younger age group (15–24 years). Nearly one-half of mothers (47.2%) had a secondary level of education. About one in every four mothers (24.6%) was underweight. The majority of the participants lived in rural areas (71.3%), belonged to the OBC category (47.3%), and were Hindu (78.7%). Most of the households were male-headed (86.5%). The proportionate share of samples decreased with higher household wealth quintiles (**Table 1**).

**Table 1. Distribution of study participants (n = 38,685), NFHS-4 (2015–16).**

| Variables | Number (n) | Percentage (%) |
|---|---|---|
| *Key Predictor* | | |
| **Women Autonomy Index (WAI)** | | |
| Low | 6,817 | 19.3 |
| Moderate | 22,593 | 58.2 |
| High | 9,275 | 22.5 |
| *Child Characteristics* | | |
| **Children's age in months (mean/SD)** | | 29.9 (17.0) |
| 0–11 | 7,246 | 18.7 |
| 12–35 | 15,638 | 40.6 |

(*Continued*)

**Table 1.** (Continued)

| Variables | Number (n) | Percentage (%) |
|---|---|---|
| 36–59 | 15,801 | 40.8 |
| **Sex of children** | | |
| Male | 19,949 | 51.2 |
| Female | 18,736 | 48.8 |
| **Birth order** | | |
| <3 | 26,290 | 70.7 |
| 3+ | 12,395 | 29.3 |
| *Maternal Characteristics* | | |
| **Maternal age in years (mean/SD)** | | 27.3 (5.1) |
| 15–24 | 12,053 | 34.3 |
| 25–34 | 22,722 | 57.5 |
| 35–49 | 3,910 | 8.2 |
| **Maternal education** | | |
| No education | 11,369 | 28.2 |
| Primary | 5,439 | 13.3 |
| Secondary | 17,984 | 47.2 |
| Higher | 3,893 | 11.4 |
| **Maternal BMI** | | |
| Underweight | 8,982 | 24.6 |
| Normal | 23,747 | 59.4 |
| Overweight/Obese | 5,816 | 16.1 |
| *Household Characteristics* | | |
| **Place of residence** | | |
| Urban | 9,474 | 28.7 |
| Rural | 29,211 | 71.3 |
| **Caste** | | |
| SC | 7,100 | 21.1 |
| ST | 7,602 | 10.8 |
| OBC | 15,162 | 47.3 |
| None of them | 6,949 | 20.8 |
| **Religion** | | |
| Hindu | 27,939 | 78.7 |
| Muslim | 6,421 | 16.6 |
| Other | 4,325 | 4.7 |
| **HH size** | | |
| 0–4 | 9,393 | 25.7 |
| 5–6 | 14,051 | 36.0 |
| 6+ | 15,241 | 38.3 |
| **Sex of the HH head** | | |
| Male | 33,490 | 86.5 |
| Female | 5,195 | 13.5 |
| **Wealth index** | | |
| Poorest | 9,509 | 23.8 |
| Poorer | 8,952 | 21.7 |
| Middle | 7,918 | 20.5 |
| Richer | 6,634 | 18.1 |
| Richest | 5,672 | 15.9 |

## Child nutritional status by selected explanatory variables

Overall, 38%, 21%, and 35% of children were stunted, wasted, and underweight, respectively. The prevalence of children's stunting, wasting, and underweight decreased with higher levels of women's autonomy. We found significant differences in children's nutritional outcomes by a range of socio-demographic characteristics. The prevalence of being stunted and underweight was lower among infants (0–11 months) than among children aged 12–35 and 36–59 months old. Unlike stunting and underweight, the prevalence of wasting decreased with age. Females and lower-order (below third order) children were observed to be better nourished than their counterparts. Stunting and underweight were found to be more prevalent among children who were born to mothers aged 35–49 years than the younger ones. The prevalence of childhood malnutrition significantly decreased with increasing levels of maternal education. A significantly higher proportion of underweight mothers had stunted, wasted, and underweight children than normal and overweight/obese women. Childhood malnutrition was common among children who resided in rural areas, belonged to socially marginalized castes (SC/ST), and those from poor economic backgrounds (Table 2).

**Table 2. Prevalence of stunting, wasting, and underweight among under-five children by selected explanatory variables, NFHS-4 (2015–16).**

| Variables | Stunting | | Wasting | | Underweight | |
|---|---|---|---|---|---|---|
| | % | *p* value | % | *p* value | % | *p* value |
| *Key Predictor* | | | | | | |
| **Women Autonomy Index (WAI)** | | <0.001 | | <0.001 | | <0.001 |
| Low | 39.5 | | 22.9 | | 37.7 | |
| Moderate | 38.1 | | 20.7 | | 34.8 | |
| High | 34.9 | | 19.9 | | 32.9 | |
| *Child Characteristics* | | | | | | |
| **Children's age (months)** | | <0.001 | | <0.001 | | <0.001 |
| 0–11 | 21.0 | | 29.0 | | 26.3 | |
| 12–35 | 42.0 | | 20.1 | | 35.9 | |
| 36–59 | 40.9 | | 18.2 | | 37.9 | |
| **Sex of children** | | 0.001 | | <0.001 | | 0.034 |
| Male | 38.2 | | 22.1 | | 35.2 | |
| Female | 37.0 | | 19.7 | | 34.6 | |
| **Birth order** | | <0.001 | | <0.001 | | <0.001 |
| <3 | 34.4 | | 20.4 | | 31.9 | |
| 3+ | 45.5 | | 22.2 | | 42.3 | |
| *Maternal Characteristics* | | | | | | |
| **Maternal age (years)** | | <0.001 | | <0.001 | | 0.070 |
| 15–24 | 37.1 | | 21.7 | | 34.5 | |
| 25–34 | 37.2 | | 20.4 | | 34.5 | |
| 35–49 | 43.2 | | 21.1 | | 39.9 | |
| **Maternal education** | | <0.001 | | <0.001 | | <0.001 |
| No education | 50.7 | | 23.3 | | 47.5 | |
| Primary | 42.9 | | 22.2 | | 39.8 | |
| Secondary | 32.9 | | 19.8 | | 30.3 | |
| Higher | 19.0 | | 18.3 | | 17.5 | |
| **Maternal BMI** | | <0.001 | | <0.001 | | <0.001 |
| Underweight | 45.3 | | 27.0 | | 48.3 | |
| Normal | 37.7 | | 19.9 | | 33.4 | |

*(Continued)*

**Table 2.** (Continued)

| Variables | Stunting | | Wasting | | Underweight | |
|---|---|---|---|---|---|---|
| | % | *p* value | % | *p* value | % | *p* value |
| Overweight/Obese | 25.9 | | 15.1 | | 19.9 | |
| *Household Characteristics* | | | | | | |
| **Place of residence** | | <0.001 | | 0.006 | | <0.001 |
| Urban | 28.7 | | 20.6 | | 27.8 | |
| Rural | 41.2 | | 21.1 | | 37.8 | |
| **Caste** | | <0.001 | | <0.001 | | <0.001 |
| SC | 43.7 | | 21.1 | | 39.3 | |
| ST | 44.9 | | 27.0 | | 45.2 | |
| OBC | 37.1 | | 20.6 | | 35.2 | |
| None of them | 29.9 | | 19.4 | | 26.6 | |
| **Religion** | | <0.001 | | <0.001 | | <0.001 |
| Hindu | 37.9 | | 21.6 | | 35.8 | |
| Muslim | 38.5 | | 18.4 | | 32.8 | |
| Other | 30.9 | | 18.0 | | 28.0 | |
| **HH size** | | <0.001 | | 0.961 | | <0.001 |
| 0–4 | 35.6 | | 21.2 | | 32.9 | |
| 5–6 | 37.9 | | 20.7 | | 35.1 | |
| 6+ | 38.8 | | 20.9 | | 36.2 | |
| **Sex of the HH head** | | 0.273 | | 0.178 | | 0.626 |
| Male | 37.3 | | 20.9 | | 34.7 | |
| Female | 39.7 | | 20.9 | | 36.3 | |
| **Wealth index** | | <0.001 | | <0.001 | | <0.001 |
| Poorest | 51.5 | | 25.0 | | 49.5 | |
| Poorer | 44.7 | | 21.0 | | 40.6 | |
| Middle | 35.5 | | 19.9 | | 32.2 | |
| Richer | 28.2 | | 19.0 | | 25.9 | |
| Richest | 20.8 | | 18.2 | | 19.1 | |
| **Overall** | **37.6** | | **20.9** | | **34.9** | |

Note: P values are derived from Pearson's Chi-square test of association between outcome variables and explanatory variables.

### Association between maternal autonomy and child nutritional outcomes

We employed stepwise logistic regression models for assessing the association between maternal autonomy and stunting (**Table 3**), wasting (**Table 4**), and underweight (**Table 5**) in children. The pairwise analysis (Model 1) indicates a statistically significant association between women's autonomy and children's nutritional outcomes, where children of mothers with high autonomy had lower odds of being stunted (OR: 0.77; 95% CI 0.73, 0.83), wasted (OR: 0.80; 95% CI 0.74, 0.86), and underweight (OR: 0.72; 95% CI 0.67, 0.77) compared to women having low autonomy. The magnitude of this association remains relatively unchanged in model 2 upon the addition of child characteristics, whereas the magnitude of the association gets progressively weaker upon the inclusion of other explanatory variables (maternal and household characteristics) subsequently in models 3 and 4 (though it remained statistically significant). The association between maternal autonomy and children's stunting and wasting became statistically insignificant in the full model (in model 5), which potentially suggests a greater influence of confounding variables such as child, maternal, and household characteristics.

**Table 3. Binary logistic regression models assessing the association between maternal autonomy and stunting among under-five children, NFHS-4 (2015–16).**

| Stunting | Model 1: OR (95% CI) | Model 2: OR (95% CI) | Model 3: OR (95% CI) | Model 4: OR (95% CI) | Model 5: OR (95% CI) |
|---|---|---|---|---|---|
| *Key Predictor* | | | | | |
| **Women Autonomy Index (WAI)** | | | | | |
| Low (ref.) | | | | | |
| Moderate | 0.89 (0.84, 0.94)** | 0.86 (0.81, 0.91)** | 0.94 (0.89, 1.00)* | 0.91 (0.86, 0.97)** | 0.95 (0.90, 1.01) |
| High | 0.77 (0.73, 0.83)** | 0.73 (0.68, 0.78)** | 0.91 (0.85, 0.97)** | 0.86 (0.80, 0.92)** | 0.93 (0.87, 1.00) |
| *Child Characteristics* | | | | | |
| **Children's age (months)** | | | | | |
| 0–11 (ref.) | | | | | |
| 12–35 | | 2.63 (2.47, 2.81)** | 2.72 (2.55, 2.90)** | 2.79 (2.61, 2.98)** | 2.81 (2.63, 3.01)** |
| 36–59 | | 2.54 (2.38, 2.71)** | 2.66 (2.49, 2.85)** | 2.61 (2.44, 2.79)** | 2.69 (2.51, 2.88)** |
| **Sex of children** | | | | | |
| Male (ref.) | | | | | |
| Female | | 0.93 (0.90, 0.97)** | 0.92 (0.88, 0.96)** | 0.92 (0.88, 0.96)** | 0.92 (0.88, 0.96)** |
| **Birth order** | | | | | |
| <3 (ref.) | | | | | |
| 3+ | | 1.51 (1.45, 1.58)** | 1.29 (1.22, 1.36)** | 1.17 (1.11, 1.23)** | 1.17 (1.11, 1.24)** |
| *Maternal Characteristics* | | | | | |
| **Maternal age (years)** | | | | | |
| 15–24 (ref.) | | | | | |
| 25–34 | | | 0.81 (0.77, 0.86)** | | 0.87 (0.82, 0.91)** |
| 35–49 | | | 0.77 (0.71, 0.84)** | | 0.83 (0.76, 0.91)** |
| **Maternal education** | | | | | |
| No education (ref.) | | | | | |
| Primary | | | 0.79 (0.74, 0.85)** | | 0.89 (0.83, 0.95)** |
| Secondary | | | 0.57 (0.54, 0.60)** | | 0.73 (0.69, 78)** |
| Higher | | | 0.34 (0.31, 0.37)** | | 0.54 (0.48, 0.60)** |
| *Maternal BMI* | | | | | |
| Underweight | | | 1.31 (1.24, 1.37)** | | 1.23 (1.17, 1.30)** |
| Normal (ref.) | | | | | |
| Overweight/Obese | | | 0.66 (0.62, 0.71)** | | 0.78 (0.72, 0.83)** |
| *Household Characteristics* | | | | | |
| **Place of residence** | | | | | |
| Urban (ref.) | | | | | |
| Rural | | | | 0.96 (0.90, 1.02) | 0.94 (0.89, 1.00) |
| **Caste** | | | | | |
| SC | | | | 1.41 (1.31, 1.53)** | 1.34 (1.24, 1.44)** |
| ST | | | | 1.21 (1.12, 1.31)** | 1.14 (1.06, 1.24)** |
| OBC | | | | 1.24 (1.16, 1.32)** | 1.19 (1.11, 1.27)** |
| None of them (ref.) | | | | | |
| **Religion** | | | | | |
| Hindu (ref.) | | | | | |
| Muslim | | | | 1.16 (1.09, 1.24)** | 1.12 (1.05, 1.20)** |
| Other | | | | 0.84 (0.77, 0.91)** | 0.89 (0.82, 97)** |
| **HH size** | | | | | |
| 0–4 (ref.) | | | | | |
| 5–6 | | | | 1.02 (0.96, 1.08) | 1.01 (0.95, 1.07) |
| 6+ | | | | 1.13 (1.06, 1.20)** | 1.10 (1.04, 1.17)** |

*(Continued)*

**Table 3.** (Continued)

| Stunting | Model 1: OR (95% CI) | Model 2: OR (95% CI) | Model 3: OR (95% CI) | Model 4: OR (95% CI) | Model 5: OR (95% CI) |
|---|---|---|---|---|---|
| **Sex of the HH head** | | | | | |
| Male (ref.) | | | | | |
| Female | | | | 1.03 (0.96, 1.10) | 1.03 (0.96, 1.10) |
| **Wealth index** | | | | | |
| Poorest (ref.) | | | | | |
| Poorer | | | | 0.76 (0.71, 0.81)** | 0.83 (0.78, 0.88)** |
| Middle | | | | 0.54 (0.50, 0.57)** | 0.63 (0.59, 0.68)** |
| Richer | | | | 0.40 (0.37, 0.43)** | 0.51 (0.47, 56)** |
| Richest | | | | 0.28 (0.26, 0.31)** | 0.43 (0.39, 0.47)** |

Significance level

**$p<0.01$

*$p<0.05$.

Abbreviation: OR: Odds ratio, CI: Confidence interval; ref.: Reference category.

However, maternal autonomy retained its significant relationship (though marginally) with underweight even after adjusting for controlling factors in the full model, where children of mothers with moderate autonomy were 6% less likely to be underweight (OR: 0.94; 95% CI: 0.88, 0.99) than those who had low autonomy.

Child demographics (i.e., age, sex, and birth order) were significantly associated with nutritional indicators. The odds of being stunted and underweight were higher among children aged 12–35 and 36–59 months than among infants (0–11 months). Females had lower chances of being stunted, wasted, and underweight than male children. Maternal education had an inverse relationship with childhood malnutrition, indicating a decrease in the probability of stunting, wasting, and being underweight with increasing levels of education. Children of underweight mothers were more likely to be malnourished than mothers who had a normal BMI. Among household characteristics, the household wealth quintile had a strong negative correlation with childhood malnutrition, where the odds of stunting, wasting, and being underweight were reduced with upper household wealth quintiles.

## Discussion

The findings of this study indicate that maternal autonomy is inextricably associated with children's nutritional status, where higher levels of maternal autonomy are associated with lower odds of stunting, wasting, and being underweight among under-five children to a certain extent. However, the strength of the association became weak when we controlled for all confounding variables (child, maternal, and household characteristics) in the full model that indicates the intrinsic association between maternal autonomy and childhood malnutrition. The possible reason for this weak and marginal relationship could be attributed to the inclusion of a greater number of confounding factors in the full model. Our study also suggests that variations in maternal education, BMI of mothers, and household wealth quintile primarily explained children's malnutrition in India. Prior studies conducted in India [17, 18], Bangladesh [26], and Pakistan [23] show that women's greater degree of autonomy/agency is significantly correlated better nutritional outcomes of children. Women with greater freedom of movement can step out into neighbourhood places and proximal markets, which increases the chances of being exposed to health-related knowledge that could be beneficial for the health and nutrition of children [9, 18]. Household decision-making power enables women to

**Table 4. Binary logistic regression models assessing the association between maternal autonomy and wasting among under-five children, NFHS-4 (2015–16).**

| Wasting | Model 1: OR (95% CI) | Model 2: OR (95% CI) | Model 3: OR (95% CI) | Model 4: OR (95% CI) | Model 5: OR (95% CI) |
|---|---|---|---|---|---|
| *Key Predictor* | | | | | |
| **Women Autonomy Index (WAI)** | | | | | |
| Low (ref.) | | | | | |
| Moderate | 0.88 (0.83, 0.94)** | 0.89 (0.84, 0.95)** | 0.94 (0.88, 1.00) | 0.92 (0.86, 0.98)* | 0.94 (0.87, 1.00) |
| High | 0.80 (0.74, 0.86)** | 0.82 (0.76, 0.89)** | 0.91 (0.84, 0.99)* | 0.89 (0.82, 0.96)** | 0.92 (0.85, 1.00) |
| *Child Characteristics* | | | | | |
| **Children's age (months)** | | | | | |
| 0–11 (ref.) | | | | | |
| 12–35 | | 0.65 (0.61, 0.69)** | 0.64 (0.60, 0.68)** | 0.65 (0.61, 0.69)** | 0.64 (0.60, 0.68)** |
| 36–59 | | 0.56 (0.53, 0.60)** | 0.55 (0.52, 0.59)** | 0.55 (0.52, 0.59)** | 0.55 (0.51, 0.59)** |
| **Sex of children** | | | | | |
| Male (ref.) | | | | | |
| Female | | 0.90 (0.85, 0.94)** | 0.89 (0.84, 0.93)** | 0.89 (0.84, 0.94)** | 0.89 (0.84, 0.93)** |
| **Birth order** | | | | | |
| <3 (ref.) | | | | | |
| 3+ | | 1.12 (1.07, 1.18)** | 1.04 (0.98, 1.11) | 1.03 (0.97, 1.09) | 1.02 (0.95, 1.09) |
| *Maternal Characteristics* | | | | | |
| **Maternal age (years)** | | | | | |
| 15–24 (ref.) | | | | | |
| 25–34 | | | 1.01 (0.95, 1.07) | | 1.03 (0.97, 1.10) |
| 35–49 | | | 0.94 (0.85, 1.04) | | 1.00 (0.90, 1.11) |
| **Maternal education** | | | | | |
| No education (ref.) | | | | | |
| Primary | | | 0.88 (0.81, 0.95)** | | 0.95 (0.87, 1.03) |
| Secondary | | | 0.81 (0.76, 0.86)** | | 0.91 (0.84, 0.97)** |
| Higher | | | 0.75 (0.67, 0.83)** | | 0.85 (0.76, 0.96)** |
| **Maternal BMI** | | | | | |
| Underweight | | | 1.54 (1.45, 1.63)** | | 1.46 (1.38, 1.55)** |
| Normal (ref.) | | | | | |
| Overweight/Obese | | | 0.69 (0.64, 0.75)** | | 0.74 (0.68, 0.81)** |
| *Household Characteristics* | | | | | |
| **Place of residence** | | | | | |
| Urban (ref.) | | | | | |
| Rural | | | | 0.87 (0.82, 0.94)** | 0.86 (0.80, 0.93)** |
| **Caste** | | | | | |
| SC | | | | 1.16 (1.06, 1.27)** | 1.14 (1.04, 1.25)** |
| ST | | | | 1.36 (1.24, 1.49)** | 1.33 (1.21, 1.46)** |
| OBC | | | | 1.14 (1.06, 1.23)** | 1.13 (1.05, 1.22)** |
| None of them (ref.) | | | | | |
| **Religion** | | | | | |
| Hindu (ref.) | | | | | |
| Muslim | | | | 0.85 (0.79, 0.92)** | 0.86 (0.80, 0.94)** |
| Other | | | | 0.59 (0.53, 0.65)** | 0.63 (0.57, 0.70)** |
| **HH size** | | | | | |
| 0–4 (ref.) | | | | | |
| 5–6 | | | | 0.97 (0.90, 1.04) | 0.96 (0.89, 1.03) |
| 6+ | | | | 0.97 (0.90, 1.04) | 0.95 (0.89, 1.02) |

*(Continued)*

**Table 4.** (Continued)

| Wasting | Model 1: OR (95% CI) | Model 2: OR (95% CI) | Model 3: OR (95% CI) | Model 4: OR (95% CI) | Model 5: OR (95% CI) |
|---|---|---|---|---|---|
| **Sex of the HH head** | | | | | |
| Male (ref.) | | | | | |
| Female | | | | 0.93 (0.86, 1.01) | 0.93 (0.86, 1.00)* |
| **Wealth index** | | | | | |
| Poorest (ref.) | | | | | |
| Poorer | | | | 0.78 (0.73, 0.84)** | 0.83 (0.77, 0.89)** |
| Middle | | | | 0.71 (0.65, 0.76)** | 0.78 (0.72, 0.85)** |
| Richer | | | | 0.66 (0.60, 0.72)** | 0.78 (0.71, 0.86)** |
| Richest | | | | 0.59 (0.53, 0.65)** | 0.74 (0.66, 0.83)** |

Significance level

**$p < 0.01$

*$p < 0.05$.

Abbreviation: OR: Odds ratio, CI: Confidence interval; ref.: Reference category.

directly participate in and control household purchasing choices, which allows them to selectively purchase nutritious inputs, including nutritious foods for children's growth and development [17, 22, 26, 44–47]. Access to financial resources or control over the assets of mothers may also lead to better nourishment for their children. Studies indicate that mothers who have control over material resources, including money in hand, may invest more in their children's healthcare and nutritious foods [42].

With regard to maternal characteristics, education is an important aspect of agency for mothers that may have a beneficial influence on children's health and nutrition. Similar to a study by Shroff et al. [17], our findings also exhibited a strong and independent association between maternal education and child nutritional status, indicating an increasing level of maternal education significantly reduces the risk of childhood malnutrition. Education enables women to acquire proper healthcare-related information and establishes individual agencies regarding healthcare access for them as well as their children, which eventually leads to better nourishment for children [26, 30, 48, 49]. We have found that underweight mothers (BMI <18.5 kg/m$^2$) tend to have more malnourished children. This finding corroborates with Rahman et al. [26], Yaya et al. [29], and Jones et al. [50], indicating that underweight women may have insufficient breast milk production and inadequate/deficient biological transfer of nutrition to the foetus. Moreover, food insecurity due to financial hardship leads to dysregulated dietary patterns for both mothers and children, which directly affects the physical and emotional well-being of children. The fundamental pathway of women's autonomy and childhood nutrition is mediated by women's nutritional health, which is controlled by their engagement in decision-making power in the household [50]. Our study demonstrated maternal age to be a strong predictor of childhood nutritional outcomes even after controlling for all explanatory variables in the analysis. Our findings are similar to other studies [51, 52] that suggest young mothers are highly susceptible to having malnourished children compared to older mothers. Marriage at younger ages is associated with lower levels of education and lack of empowerment, thus reducing women's self-esteem and controlling power over household decision-making, which hinders successfully securing children's health [51, 53, 54]. Studies also indicate that young mothers (with insufficient iron concentration) are prone to delivering premature and low birth weight babies, possibly explaining future malnourishment in these populations [51]. Children living in rural settings were found to have a significant independent association

**Table 5. Binary logistic regression models assessing the association between maternal autonomy and underweight among under-five children, NFHS-4 (2015–16).**

| Underweight | Model 1: OR (95% CI) | Model 2: OR (95% CI) | Model 3: OR (95% CI) | Model 4: OR (95% CI) | Model 5: OR (95% CI) |
|---|---|---|---|---|---|
| *Key Predictor* | | | | | |
| **Women Autonomy Index (WAI)** | | | | | |
| Low (ref.) | | | | | |
| Moderate | 0.84 (0.80, 0.89)** | 0.83 (0.78, 0.87)** | 0.92 (0.86, 0.97)** | 0.89 (0.84, 0.94)** | 0.94 (0.88, 0.99)* |
| High | 0.72 (0.67, 0.77)** | 0.69 (0.64, 0.74)** | 0.88 (0.82, 0.94)** | 0.84 (0.79, 0.91)** | 0.93 (0.87, 1.00) |
| *Child Characteristics* | | | | | |
| **Children's age (months)** | | | | | |
| 0–11 (ref.) | | | | | |
| 12–35 | | 1.50 (1.41, 1.59)** | 1.51 (1.41, 1.61)** | 1.55 (1.45, 1.65)** | 1.54 (1.44, 1.64)** |
| 36–59 | | 1.63 (1.53, 1.73)** | 1.68 (1.57, 1.79)** | 1.64 (1.54, 1.75)** | 1.68 (1.57, 1.80)** |
| **Sex of children** | | | | | |
| Male (ref.) | | | | | |
| Female | | 0.96 (0.92, 1.00) | 0.95 (0.91, 0.99)* | 0.95 (0.90, 0.99)* | 0.94 (0.90, 0.99)* |
| **Birth order** | | | | | |
| <3 (ref.) | | | | | |
| 3+ | | 1.48 (1.42, 1.55)** | 1.25 (1.18, 1.32)** | 1.17 (1.11, 1.23)** | 1.16 (1.09, 1.23)** |
| *Maternal Characteristics* | | | | | |
| **Maternal age (years)** | | | | | |
| 15–24 (ref.) | | | | | |
| 25–34 | | | 0.88 (0.83, 0.92)** | | 0.93 (0.88, 0.98)* |
| 35–49 | | | 0.77 (0.70, 0.84)** | | 0.85 (0.77, 0.93)** |
| **Maternal education** | | | | | |
| No education (ref.) | | | | | |
| Primary | | | 0.76 (0.71, 0.81)** | | 0.87 (0.81, 0.94)** |
| Secondary | | | 0.56 (0.53, 0.59)** | | 0.73 (0.69, 0.78)** |
| Higher | | | 0.34 (0.31, 0.37)** | | 0.52 (0.46, 0.58)** |
| **Maternal BMI** | | | | | |
| Underweight | | | 1.79 (1.70, 1.88)** | | 1.63 (1.55, 1.72)** |
| Normal (ref.) | | | | | |
| Overweight/Obese | | | 0.53 (0.49, 0.57)** | | 0.62 (0.58, 0.67)** |
| *Household Characteristics* | | | | | |
| **Place of residence** | | | | | |
| Urban (ref.) | | | | | |
| Rural | | | | 0.87 (0.82, 0.92)** | 0.85 (0.79, 0.90)** |
| **Caste** | | | | | |
| SC | | | | 1.47 (1.36, 1.59)** | 1.39 (1.28, 1.50)** |
| ST | | | | 1.34 (1.24, 1.46)** | 1.25 (1.15, 1.36)** |
| OBC | | | | 1.36 (1.27, 1.45)** | 1.31 (1.22, 1.40)** |
| None of them (ref.) | | | | | |
| **Religion** | | | | | |
| Hindu (ref.) | | | | | |
| Muslim | | | | 0.96 (0.90, 1.03) | 0.95 (0.88, 1.02) |
| Other | | | | 0.54 (0.50, 0.59)** | 0.61 (0.56, 0.66)** |
| **HH size** | | | | | |
| 0–4 (ref.) | | | | | |
| 5–6 | | | | 1.02 (0.96, 1.08) | 1.00 (0.94, 1.07) |
| 6+ | | | | 1.09 (1.03, 1.16)** | 1.06 (1.00, 1.13) |

*(Continued)*

**Table 5.** (Continued)

| Underweight | Model 1: OR (95% CI) | Model 2: OR (95% CI) | Model 3: OR (95% CI) | Model 4: OR (95% CI) | Model 5: OR (95% CI) |
|---|---|---|---|---|---|
| **Sex of the HH head** | | | | | |
| Male (ref.) | | | | | |
| Female | | | | 1.00 (0.94, 1.07) | 0.99 (0.93, 1.06) |
| **Wealth index** | | | | | |
| Poorest (ref.) | | | | | |
| Poorer | | | | 0.68 (0.64, 0.72)** | 0.76 (0.71, 0.81)** |
| Middle | | | | 0.49 (0.46, 0.53)** | 0.60 (0.56, 0.65)** |
| Richer | | | | 0.37 (0.34, 0.40)** | 0.52 (0.48, 0.57)** |
| Richest | | | | 0.26 (0.23, 0.28)** | 0.43 (0.39, 0.48)** |

Significance level

**$p<0.01$

*$p<0.05$.

Abbreviation: OR: Odds ratio, CI: Confidence interval; ref.: Reference category.

with an increased likelihood of being wasted and underweight, which could be explained by high levels of poverty, the persistent prevalence of child marriage associated with early pregnancy and lower educational attainment among women in those settings [51, 55]. Children from well-off families have improved nutritional outcomes (lower risk of stunting, wasting, and being underweight) compared to poor families due to the availability of consistently diverse food and high-quality nutrient supplements in wealthier households [30, 50]. Moreover, earlier studies have highlighted a complex nexus between women's autonomy, household wealth, and childhood nutrition with possible effects modified by contextual differences (availability of infrastructure/health resources in particular settings) [50]. Evidence suggests that better household wealth status does not necessarily establish individual agency among women as patriarchy prevails with the predominant male dominance of the breadwinner [50, 56]. Owing to low socioeconomic status (SES), women from marginalized communities are often forced to step out for income generation, which may have a positive impact on their degree of freedom. On the other hand, poor SES is associated with lower cognitive development in children, resulting in long-term poor productivity and social and economic well-being [51].

## Strengths and limitations

To the best of our knowledge, our study findings are instrumental in understanding the association between women's autonomy and the nutritional status of children in India. We used nationally representative samples in all our measurements; therefore, the results have strong external validity and could be generalised to the whole country. Moreover, the inclusion of a wide range of confounding factors yields robust and consistent results. Our study findings have significant value in combating childhood malnutrition by promoting women's decision-making power, freedom of movement, and access to financial resources/control over assets. Despite many important findings, our study has several limitations. Due to the cross-sectional nature of the data, we failed to establish a causal relationship between maternal autonomy and child nutrition. Given the retrospective study design and self-reported data, our findings are prone to recall bias and social desirability bias. Since women's autonomy is a multifaceted aspect, its measurement is complex and may not be captured completely through the three dimensions included in this study. Another limitation of this study was identifying pathways through which maternal autonomy is associated with child nutritional status, which requires

path or mediation analysis. Furthermore, the nutritional status of children is not always necessarily explained only by our included socio-demographic characteristics. Several other factors such as utilization of maternity care services, family planning and personal childcare practices, birth size, mother-child dietary patterns, past disease exposures, environmental health factors, and access to proximal health services are also important in determining childhood malnutrition, which were not included in this study [50].

## Conclusions

Our study findings indicate that maternal autonomy is marginally associated with stunting, wasting, and underweight in children under five. In particular, only underweight has retained its significant relationship after controlling for all confounding factors in the model. Furthermore, we observed that maternal education, maternal BMI, and household wealth have a profound influence on the nutritional status of children. Our findings do not only inform women's empowerment programs but also reinforce effective interventions towards improving female educational attainment and nutritional status of women as well as addressing socio-economic inequalities in order to combat the persistent burden of childhood malnutrition in India.

## Author Contributions

**Conceptualization:** Pintu Paul, Ria Saha.

**Data curation:** Pintu Paul.

**Formal analysis:** Pintu Paul, Ria Saha.

**Investigation:** Pintu Paul, Ria Saha.

**Methodology:** Pintu Paul, Ria Saha.

**Software:** Pintu Paul.

**Supervision:** Pintu Paul.

**Validation:** Pintu Paul, Ria Saha.

**Visualization:** Pintu Paul, Ria Saha.

**Writing – original draft:** Pintu Paul, Ria Saha.

**Writing – review & editing:** Pintu Paul, Ria Saha.

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
