## [Decision Letter · Decision Letter 0]

28 Feb 2022

PONE-D-21-31159Maternal autonomy and child nutritional status in India: Evidence from a nationally representative population-based surveyPLOS ONE

Dear Dr. Paul,

Thank you for submitting your manuscript to PLOS ONE. After careful consideration, we feel that it has merit but does not fully meet PLOS ONE’s publication criteria as it currently stands. Therefore, we invite you to submit a revised version of the manuscript that addresses the points raised during the review process.

We look forward to receiving your revised manuscript.

Kind regards,

Kannan Navaneetham, PhD

Academic Editor

PLOS ONE

https://journals.plos.org/plosone/s/fileid=ba62/PLOSOne_formatting_sample_title_authors_affiliations.pdf".

Reviewers' comments:

Reviewer's Responses to Questions

**Comments to the Author**

1. Is the manuscript technically sound, and do the data support the conclusions?

Reviewer #1: Yes

Reviewer #2: Partly

2. Has the statistical analysis been performed appropriately and rigorously? 

Reviewer #1: Yes

Reviewer #2: Yes

3. Have the authors made all data underlying the findings in their manuscript fully available?

Reviewer #1: No

Reviewer #2: Yes

4. Is the manuscript presented in an intelligible fashion and written in standard English?

Reviewer #1: No

Reviewer #2: Yes

5. Review Comments to the Author

Reviewer #1: This cross-sectional study investigated the association between maternal autonomy and child nutritional status in India using the nationally representative population-based survey. In general, the study topic is an important issue for child nutrition but not a novel idea. Here are some comments for your consideration:

Abstract: Although the author has described the general findings in the abstract, I suggest adding the data for the result implication.

Introduction:

Line 73-76: the references to support the description are needed.

Materials and methods

Line 191: “a total of 45,231 mother-child pairs were available in the module.” However, 6546 pairs (14%) are excluded for some reasons. Is the rest sample still represent the original population? On the other hand, is any siblings included in the analysis sample? Or do all paired come from different households?

Line 210: the format of citation is incorrect.

Line 237: how to decide the cut-off points and the distribution of each dimension for women autonomy index (WAI).

Line 238: the description of confounding variables is too redundant. I suggest the author to simplified.

Line 263: did the author test for interaction among the independent variables?

Line 276: please provide the VIF value.

Results

Line 337-355: the description of confounding variables is too redundant. I suggest the author to simplified.

Discussion

Line 361-362: “The possible reason for this weak and marginal relationship could be attributed to the inclusion of a greater number of covariates in the present study.” It is necessary to discuss deeper which covariate to mediate the association between maternal autonomy and child nutrition.

Line 416-417: “Our findings are the first of a kind and instrumental in understanding the complex pathways between women’s autonomy and childhood nutrition in India.” This sentence is too strong and over-express the results of the study.

Table 3-5: Two model 3 and without model 5 in the table.

Reviewer #2: Thank you for the opportunity to review this paper assessing women’s autonomy and children’s nutritional status in India. It’s generally well written and reports some interesting results. However, the clarity of the paper can be improved substantially. My main concerns detailed below are around the methods and lack of clarity there on a number of points. I would also caution the authors against using causal language given the cross-sectional nature of their data.

Abstract

Line 45: Please define maternal autonomy in the abstract. Different indicators are used with NFHS/DHS type of data.

Lines 55-58: The conclusion is not substantiated by the results presented in the abstract. It’s quite a jump from maternal autonomy to nutrition-specific interventions, etc. I expected to see a conclusion relating to autonomy.

Introduction

Lines 67-76: Background needs to be properly referenced.

Lines 98-100: If I remember correctly, Smith et al. did not find significant associations for all nutritional status indicators and did find heterogeneity across regions. It would be worth noting such details to better contextualise your study and findings. Did Smith et al. include India?

Lines 120-123: None of these studies are causal. I suggest the authors tone down the causal language.

Lines 124-140: Can be condensed. Since you focus on India, I suggest including less detail on studies in Africa and expanding on the studies in Pakistan, Nepal, and Bangladesh.

Lines 157-158: The issue with inconsistent and inconclusive results is both theoretical and empirical, and goes far beyond what covariates are included. I suggest either dropping this sentences or expanding on other methodological limitations of existing studies and how those are being addressed in the present study.

Methods

Study participants: decision-making questions are only asked of women who are married or co-habitating with a partner. Did you restrict your data to married/cohabitating women? Or did you impute decision-making for them? It’s unclear how this was handled. The 540 cases with missing data were missing data on any or all autonomy questions?

Outcome variables: is there are a reason you did not consider the continuous Z-scores and only focused on the binary indicators? This section also needs proper referencing.

Key predictor: you outlined 3 definitions of autonomy in the introduction. Which one did you adopt here?

Decision-making: variables are equal to 1 if women participated alone or jointly. Scholars have argued that joint decision making is disguised male decision-making and may not reflect empowerment. Did you conduct sensitivity analyses with this definition, e.g., variable =1 if women alone make the decision? How do your results change?

I wouldn’t say “financial independence” is common. It’s usually called something like “economy empowerment” or “access to resources”. It’s worth noting that the domain labels differ in the literature but essentially capture the same aspects of women’s empowerment. Given the inidcators you included, I think this domain is better labeled as “access to resources” or “control over assets”. Owning land and/or house doesn’t mean women are financially independent. The might still be disempowered with respect to making decisions about these assets. They are also highly illiquid.

In DHS, there are other variables on decision-making (e.g., use of own income, use of husband’s income), mobility, and assets. As far as I know, these are the same across all DHS including NFHS? Is there are a reason you did not use them? How did you arrive at/select the indicators included here?

Is the WAI something you developed here or has it been used before by others? By combining the items, you mean you summed them? How did you decide on the 3 categories of low, moderate and high?

Statistical analysis

Are descriptives weighted for representativeness?

Do the models apply weights and cluster variables?

Results

Table 1: please add proportion of stunted, wasted, and underweight children in the full sample. Or add to the text with Table 2. It’s not clear right now what proportion of children in your full sample were stunted, wasted, underweight

Lines 302-304: these are just descirptives, right? Not associations, and status does not “improve”. Lines 314: these aren’t correlations either. You either need to revise the methods of this is in fact what you did. Or revise the results and be more mindful of the language you use. This section can also substantially be cut down since the text largely repeats what is in the table.

Lines 337-355: this section can also be condensed by highlight a few of the most important significant associations

Did you assess associations with the summary autonomy score?

Discussion

Line 358: “higher levels of maternal autonomy reduce the risk..” should be “higher levels of maternal autonomy are associated with lower risk”. The causal language is unwarranted in this cross-sectional study

Lines 361-363: just including more covariates doesn’t mean that the relationship will be attenuated. You are including important confounders that help explain the relation.

Lines 363-373: I believe most of this evidence shows that more autonomous/empowered women are more likely to purchase nutritional inputs, including nutritious foods for their child, not supplements.

Lines 382-385: this will only explain malnutrition in children who are still breastfeeding. But your sample includes children 2-5 years of age who are no longer breastfed.

Strengths and limitations

Lines 416-418: with all due respect, your analysis does not help understand the pathways between women’s empowerment and child nutrition. You are simply showing they are associated and that maternal and household factors explain away some of the association. Mediation or path analyses would be needed to elucidate pathways. Plus, the only factor you consider that could be plausibly on the pathway from women’s empowerment to child nutrition is maternal nutritional status, which as you cite has previously been examined by Jones et al., albeit in East Africa.

You did use nationally representative data, but it’s not clear if your descriptives and model estimates were weighted for representativeness. If not, then you cannot generalize to all of India.

Lines 426-428: some (including myself) will argue that there are universal aspects of women’s empowerment that are applicable to all context. More importantly, I’m more concerned about the validity of your index for all of India. It’s a large and heterogeneous country, and it’s possible that your index does not capture the context-specific aspects of women’s empowerment within India itself.

I agree there may be issues with recall, but what makes you think these would be systematic?

Decision-making questions are usually asked in the man’s DHS survey. Is this not the case in NFHS?

Lines 443-444: yet, you use causal language throughout. The cross-sectional nature also implies the possibility of reverse causality. It’s possible women are more empowered because their children are healthier.

I wouldn’t consider the fact that your data is NFHS4 as a limitation. If you are concerned about this, you can easily compare key indicators your report here with key indicators in NFHS5, which have now been published.

Conclusions

I think these go beyond what you find. Agreed women’s autonomy should be considered in future interventions, but I suggest more caution with these recommendations given that your results were not significant in the full model for stunting and wasting, and only marginally significant for underweight.

Lines 456-458: I don’t see how your findings imply that empowerment programmes should be delivered through health facilities, or that more health facilities should be established. You don’t consider any healthcare access variables in your model.

Lines 459-464: Cash transfers are not a nutrition-specific intervention. Since they fall under social protection and safety nets, they are considered a nutrition-sensitive intervention. Multisectoral interventions to promote child health and nutrition have been proposed for years, and in fact many progrmmes have become multisectoral addressing multiple risk factors. I disagree that your findings lay the groundwork for such programmes. Rather they are in line with what is already recommended and perhaps help identify additional entry points.

6. PLOS authors have the option to publish the peer review history of their article (what does this mean?). If published, this will include your full peer review and any attached files.

Reviewer #1: No

Reviewer #2: No

---

## [Author Response · Author response to Decision Letter 0]

23 Mar 2022

Responses to reviewer comments 

Reviewer 1

Abstract: 

Although the author has described the general findings in the abstract, I suggest adding the data for the result implication.

Thank you so much for your suggestion. We have added the data for the implication of our findings in the abstract.

Introduction:

1. Line 73-76: the references to support the description are needed.

Thank you very much for noticing and suggesting that. We have now provided all the required and relevant references in those indicated lines. 

Materials and methods:

Line 191: “a total of 45,231 mother-child pairs were available in the module.” However, 6546 pairs (14%) are excluded for some reasons. Is the rest sample still represent the original population? On the other hand, is any siblings included in the analysis sample? Or do all paired come from different households?

After excluding few samples, our results still represent the original population. All under-five children are included in the sample; therefore, siblings are included in the analysis.

Line 210: the format of citation is incorrect.

Thank you for noticing it. We have corrected the format of citation.

Line 237: how to decide the cut-off points and the distribution of each dimension for women autonomy index (WAI)?

The cut-off points for the index are based on equal distribution of samples for each category of women autonomy index.

Line 238: the description of confounding variables is too redundant. I suggest the author to simplified.

Thank you for your suggestion. We have succinctly written the description of confounding variables in the revised version as per your suggestion.

Line 263: did the author test for interaction among the independent variables?

Yes, we did the test for interaction among the independent variables.

Line 276: please provide the VIF value.

Now, we have provided the VIF value (1.66).

Results

Line 337-355: the description of confounding variables is too redundant. I suggest the author to simplified.

We simplified and concise the description of confounding variables in the revised manuscript.

Discussion:

1. Line 361-362: “The possible reason for this weak and marginal relationship could be attributed to the inclusion of a greater number of covariates in the present study.” It is necessary to discuss deeper which covariate to mediate the association between maternal autonomy and child nutrition.

We carried out an additional analysis to identify it. We observe that maternal education, BMI, and wealth status primarily mediate the association between maternal autonomy and child nutrition.

2. Line 416-417: “Our findings are the first of a kind and instrumental in understanding the complex pathways between women’s autonomy and childhood nutrition in India.” This sentence is too strong and over-express the results of the study.

Thank you so much for this valuable suggestion. We have already changed our language in those indicated lines. 

3. Table 3-5: Two model 3 and without model 5 in the table.

Thank you for noticing. We have now corrected that. We apologize for any inconvenience that might have occurred.

Reviewer 2

Abstract:

Line 45: Please define maternal autonomy in the abstract. Different indicators are used with NFHS/DHS type of data.

Now, we have defined maternal autonomy in the abstract.

Lines 55-58: The conclusion is not substantiated by the results presented in the abstract. It’s quite a jump from maternal autonomy to nutrition-specific interventions, etc. I expected to see a conclusion relating to autonomy.

Thank you for your suggestion. We have modified this section in line with our study findings. 

Introduction:

1. Lines 67-76: Background needs to be properly referenced.

Thank you very much for noticing and suggesting that. We have now provided all the required and relevant references in those indicated lines. 

2. Lines 98-100: If I remember correctly, Smith et al. did not find significant associations for all nutritional status indicators and did find heterogeneity across regions. It would be worth noting such details to better contextualise your study and findings. Did Smith et al. include India?

Thank you for noticing this. Smith et al., 2003 paper include regions like South Asia, Sub-Saharan Africa, and Latin America and Caribbean. India (DHS: 1998) is included in this study as a part of south Asia region. The study assessed women’s status defined as “women’s power relative to men” – we have mentioned that in our manuscript too which we believe would provide more clarity while contextualising our study. Yes, you indicated right that Smith et al., 2003’s findings are heterogeneous in nature. The findings of Latin America and Caribbean is different from both South Asia and Sub-Saharan African regions. In both South Asia and Sub-Saharan Africa, the paper found a strong association of women’s status on children’s short- and long-term nutritional status but in Latin America and Caribbean, women’s social status had only positive effect on children’s short-term nutritional status and was observed in selected households where already women’s relative decision-making power was weak. We have incorporated these in a concise manner in our manuscript. 

3. Lines 120-123: None of these studies are causal. I suggest the authors tone down the causal language.

Thank you so much for this valuable suggestion. We have changed causal language. 

4. Lines 124-140: Can be condensed. Since you focus on India, I suggest including less detail on studies in Africa and expanding on the studies in Pakistan, Nepal, and Bangladesh.

Thank you for this suggestion. We have followed your advice and had concise the description on Africa and expanded more on Pakistan, Nepal, and Bangladesh. 

5. Lines 157-158: The issue with inconsistent and inconclusive results is both theoretical and empirical, and goes far beyond what covariates are included. I suggest either dropping this sentences or expanding on other methodological limitations of existing studies and how those are being addressed in the present study.

Thank you for this valuable suggestion. We have dropped this sentence.

Methods 

Study participants: decision-making questions are only asked of women who are married or co-habitating with a partner. Did you restrict your data to married/cohabitating women? Or did you impute decision-making for them? It’s unclear how this was handled. The 540 cases with missing data were missing data on any or all autonomy questions?

We restrict our data to married/cohabitating women. Since decision-making questions are only asked married/cohabiting women, we had to drop 540 cases from all autonomy questions to construct the WAI. 

Outcome variables: is there are a reason you did not consider the continuous Z-scores and only focused on the binary indicators? This section also needs proper referencing.

Since we considered three widely used indicators of childhood malnutrition (stunting, wasting, and underweight), the continuous Z-scores were grouped into binary categories (yes/no).

Key predictor: you outlined 3 definitions of autonomy in the introduction. Which one did you adopt here?

These definitions are broad in the context of autonomy and cannot be captured through our limited data available in the NFHS. We have used three most important domains of autonomy which are also considered in the previous studies. We mentioned those studies in methods section.

Decision-making: variables are equal to 1 if women participated alone or jointly. Scholars have argued that joint decision making is disguised male decision-making and may not reflect empowerment. Did you conduct sensitivity analyses with this definition, e.g., variable =1 if women alone make the decision? How do your results change?

We followed NFHS definition for participation in decision-making (women’s participation in household decision-making alone or jointly) in our analysis (see NFHS-4 national report). To test the validity of your argument, we did sensitivity analysis and the results do not change much.

I wouldn’t say “financial independence” is common. It’s usually called something like “economy empowerment” or “access to resources”. It’s worth noting that the domain labels differ in the literature but essentially capture the same aspects of women’s empowerment. Given the inidcators you included, I think this domain is better labeled as “access to resources” or “control over assets”. Owning land and/or house doesn’t mean women are financially independent. The might still be disempowered with respect to making decisions about these assets. They are also highly illiquid.

Thank you for your suggestion. We have changed “financial independence” to “access to resources”/“control over assets”.

In DHS, there are other variables on decision-making (e.g., use of own income, use of husband’s income), mobility, and assets. As far as I know, these are the same across all DHS including NFHS? Is there are a reason you did not use them? How did you arrive at/select the indicators included here?

For selection of variables, we primarily followed NFHS definition for computing these three domains of autonomy. In NFHS-4, for decision-making, only three questions are considered and for physical mobility, three questions are there which were considered in the present study (see NFHS-4 report). Furthermore, we conducted extensive literature search to understand and select questions for constructing the autonomy index.

Is the WAI something you developed here or has it been used before by others? By combining the items, you mean you summed them? How did you decide on the 3 categories of low, moderate and high?

We have developed WAI by summed these selected questions from the three domains of autonomy. We categorised the index into three groups based on equal number of samples in each category.

Statistical analysis

Are descriptives weighted for representativeness?

Yes, descriptive statistics are weighted for representativeness.

Do the models apply weights and cluster variables?

Yes, we applied weight and cluster variables in the models.

Results 

Table 1: please add proportion of stunted, wasted, and underweight children in the full sample. Or add to the text with Table 2. It’s not clear right now what proportion of children in your full sample were stunted, wasted, underweight

We added the proportion of stunted, wasted, and underweight children.

Lines 302-304: these are just descirptives, right? Not associations, and status does not “improve”.

Now, we have changed the language.

Lines 314: these aren’t correlations either. You either need to revise the methods of this is in fact what you did. Or revise the results and be more mindful of the language you use. This section can also substantially be cut down since the text largely repeats what is in the table.

We have revised the results and used proper language. Moreover, we cut down many sentences which repeated in the tables. 

Lines 337-355: this section can also be condensed by highlight a few of the most important significant associations.

We have succinctly written this section in the revised version.

Did you assess associations with the summary autonomy score?

Yes, we assess associations with autonomy index (categorised into 3 groups: low, moderate, and high) which is based on the summary autonomy score.

Discussion 

1. Line 358: “higher levels of maternal autonomy reduce the risk..” should be “higher levels of maternal autonomy are associated with lower risk”. The causal language is unwarranted in this cross-sectional study

Thank you for this valuable suggestion. We have now changed this language to ‘The findings of this study indicate that maternal autonomy is inextricably associated with children’s nutritional status, where higher levels of maternal autonomy is associated with lower risk of stunting, wasting, and underweight among under-five children to a certain extent.’ 

2. Lines 361-363: just including more covariates doesn’t mean that the relationship will be attenuated. You are including important confounders that help explain the relation.

Thank you for this note. We have now changed our approach of explaining that to this: ‘However, the strength of the association became weak when we controlled for all confounding variables (child, maternal, and household characteristics) in the full model that indicate the inherent association of maternal autonomy with childhood malnutrition.’

3. Lines 363-373: I believe most of this evidence shows that more autonomous/empowered women are more likely to purchase nutritional inputs, including nutritious foods for their child, not supplements.

Thank you for this suggestion. We have now changed that to nutritious food rather than supplements. 

4. Lines 382-385: this will only explain malnutrition in children who are still breastfeeding. But your sample includes children 2-5 years of age who are no longer breastfed.

While it is true that our sample is of children aged 2-5 years of age, we also believe that breastfeeding during six months to one year has a long-term positive impact on children’s nutritional status as previous studies indicated. Hence, we have included those sentences. If you think it is not necessary to include in our manuscript, we can delete that later. Thank you very much. 

Studies are listed here: https://www.karger.com/Article/Abstract/442075;
https://link.springer.com/article/10.1007/BF02758565;

Strengths and Limitations:

1. Lines 416-418: with all due respect, your analysis does not help understand the pathways between women’s empowerment and child nutrition. You are simply showing they are associated and that maternal and household factors explain away some of the association. Mediation or path analyses would be needed to elucidate pathways. Plus, the only factor you consider that could be plausibly on the pathway from women’s empowerment to child nutrition is maternal nutritional status, which as you cite has previously been examined by Jones et al., albeit in East Africa.

Thank you for this valuable suggestion. We have now changed our language related to findings of the association. We agree that we overexpressed our findings previously and we are not showing any complex pathways through our findings between women’s empowerment and childhood nutrition. Now, we have revised the strengths and limitations thoroughly. 

2. You did use nationally representative data, but it’s not clear if your descriptives and model estimates were weighted for representativeness. If not, then you cannot generalize to all of India.

We have used nationally representative data. Our all estimations are weighted for representativeness.

3. Lines 426-428: some (including myself) will argue that there are universal aspects of women’s empowerment that are applicable to all context. More importantly, I’m more concerned about the validity of your index for all of India. It’s a large and heterogeneous country, and it’s possible that your index does not capture the context-specific aspects of women’s empowerment within India itself.

Thank you for raising this important aspect. We are not capturing context-specific aspects in this study; therefore, it may not be valid for all contexts in a heterogeneous country like India. We used three domains of autonomy that can be applicable to many contexts in India and other countries as well. We assessed the simple association between maternal autonomy and child nutrition using nationally representative samples (data) which represents the aggregate association between the two and can be useful for policy interventions. We have changed our language thoroughly in the revised version. 

4. I agree there may be issues with recall, but what makes you think these would be systematic?

Since we believe that it should be just recall bias, we have ignored the word ‘systematic’. 

5. Decision-making questions are usually asked in the man’s DHS survey. Is this not the case in NFHS?

We removed this limitation.

6. Lines 443-444: yet, you use causal language throughout. The cross-sectional nature also implies the possibility of reverse causality. It’s possible women are more empowered because their children are healthier.

We have ignored the causal language in the revised manuscript. We also believe that there is a possibility of reverse causality, which has not been assessed in the present study.

7. I wouldn’t consider the fact that your data is NFHS4 as a limitation. If you are concerned about this, you can easily compare key indicators your report here with key indicators in NFHS5, which have now been published.

We have deleted this sentence from the limitation section.

Conclusions: 

1. I think these go beyond what you find. Agreed women’s autonomy should be considered in future interventions, but I suggest more caution with these recommendations given that your results were not significant in the full model for stunting and wasting, and only marginally significant for underweight.

Thank you for suggestion. Yes, we agree that our findings in the full model do not indicate a strong association between autonomy and child nutritional status. Hence, when recommend future policy actions we have taken your suggestion and have recommended that future policy actions should be directed to improve maternal education, improving maternal nutrition, and addressing socio-economic vulnerabilities.

2. Lines 456-458: I don’t see how your findings imply that empowerment programmes should be delivered through health facilities, or that more health facilities should be established. You don’t consider any healthcare access variables in your model.

Thank you for raising this matter. We have deleted those lines from the conclusion section as it seems a bit derailed from our findings.

3. Lines 459-464: Cash transfers are not a nutrition-specific intervention. Since they fall under social protection and safety nets, they are considered a nutrition-sensitive intervention. Multisectoral interventions to promote child health and nutrition have been proposed for years, and in fact many progrmmes have become multisectoral addressing multiple risk factors. I disagree that your findings lay the groundwork for such programmes. Rather they are in line with what is already recommended and perhaps help identify additional entry points.

Thank you for this valuable comment. It is great to learn this. Yes, we agree with you that our study does not necessarily establish a groundwork for the recommended programs. We have substantially revised the conclusion section in line with our study findings.

---

## [Decision Letter · Decision Letter 1]

13 Apr 2022

PONE-D-21-31159R1Is maternal autonomy associated with child nutritional status? Evidence from a cross-sectional study in IndiaPLOS ONE

Dear Dr. Paul,

Thank you for submitting your manuscript to PLOS ONE. After careful consideration, we feel that it has merit but does not fully meet PLOS ONE’s publication criteria as it currently stands. Therefore, we invite you to submit a revised version of the manuscript that addresses the points raised during the review process.

We look forward to receiving your revised manuscript.

Kind regards,

Kannan Navaneetham, PhD

Academic Editor

PLOS ONE

Reviewers' comments:

Reviewer's Responses to Questions

**Comments to the Author**

1. If the authors have adequately addressed your comments raised in a previous round of review and you feel that this manuscript is now acceptable for publication, you may indicate that here to bypass the “Comments to the Author” section, enter your conflict of interest statement in the “Confidential to Editor” section, and submit your "Accept" recommendation.

Reviewer #1: (No Response)

Reviewer #2: All comments have been addressed

2. Is the manuscript technically sound, and do the data support the conclusions?

Reviewer #1: Yes

Reviewer #2: Yes

3. Has the statistical analysis been performed appropriately and rigorously? 

Reviewer #1: Yes

Reviewer #2: Yes

4. Have the authors made all data underlying the findings in their manuscript fully available?

Reviewer #1: Yes

Reviewer #2: Yes

5. Is the manuscript presented in an intelligible fashion and written in standard English?

Reviewer #1: Yes

Reviewer #2: Yes

6. Review Comments to the Author

Reviewer #1: Line 217-220: author mentioned that “After excluding few samples, our results still represent the original population.” in the response. But, I suggest the author provide the result in supplementary files to present the result for comparing the original dataset (45231 mother-child pairs) and sub-dataset (38625 mother-child pairs) to prove the representative.

Table 3-5: “Two model 3 and without model 5 in the table.”

Reviewer #2: (No Response)

7. PLOS authors have the option to publish the peer review history of their article (what does this mean?). If published, this will include your full peer review and any attached files.

Reviewer #1: No

Reviewer #2: No

---

## [Author Response · Author response to Decision Letter 1]

20 Apr 2022

Response to Reviewers:

Thank you so much for your constructive comments and suggestions for improving our paper.

Reviewer #1: 

Line 217-220: author mentioned that “After excluding few samples, our results still represent the original population.” in the response. But, I suggest the author provide the result in supplementary files to present the result for comparing the original dataset (45231 mother-child pairs) and sub-dataset (38625 mother-child pairs) to prove the representative.

Response: Our analysis is based on the state module. However, we had to exclude a few cases from that module due to missing cases in outcome variables (i.e., cases that are out of plausible limit and flagged cases which are indicated in Figure 1). Therefore, it is not possible to conduct analysis from samples of the entire state module. We believe that the exclusion of a few samples from the state module does not change the representativeness of the data.

Table 3-5: “Two model 3 and without model 5 in the table.”

Response: It has been fixed now.

Reviewer #2: (No Response)

Thank you very much for suggesting that our manuscript does not need any further revision.

---

## [Editor Report · Decision Letter 2]

25 Apr 2022

Is maternal autonomy associated with child nutritional status? Evidence from a cross-sectional study in India

PONE-D-21-31159R2

Dear Dr. Paul,

We’re pleased to inform you that your manuscript has been judged scientifically suitable for publication and will be formally accepted for publication once it meets all outstanding technical requirements.

Kind regards,

Kannan Navaneetham, PhD

Academic Editor

PLOS ONE
---

## [Editor Report · Acceptance letter]

3 May 2022

PONE-D-21-31159R2 

Is maternal autonomy associated with child nutritional status? Evidence from a cross-sectional study in India 

Dear Dr. Paul:

I'm pleased to inform you that your manuscript has been deemed suitable for publication in PLOS ONE. Congratulations! Your manuscript is now with our production department. 

Kind regards, 

on behalf of

Prof. Kannan Navaneetham 

Academic Editor

PLOS ONE